# Clinical Validation of the Greek Version of the Acute Cystitis Symptom Score (ACSS)—Part II

**DOI:** 10.3390/antibiotics10101253

**Published:** 2021-10-15

**Authors:** Konstantinos Stamatiou, Evangelia Samara, Jakhongir F. Alidjanov, Adrian M. E. Pilatz, Kurt G. Naber, Florian M. E. Wagenlehner

**Affiliations:** 1Department of Urology, Tzaneio General Hospital, 18536 Piraeus, Greece; stamatiouk@gmail.com; 2Department of Anesthesiology and Pain Management, Tzaneio General Hospital, 18536 Piraeus, Greece; gelysamara@yahoo.com; 3Department of Urology, Pediatric Urology and Andrology, Justus-Liebig University of Giessen, 35392 Giessen, Germany; Dr.Alidjanov@gmail.com (J.F.A.); Adrian.Pilatz@chiru.med.uni-giessen.de (A.M.E.P.); Florian.Wagenlehner@chiru.med.uni-giessen.de (F.M.E.W.); 4School of Medicine, Technical University of Munich, 81664 Munich, Germany

**Keywords:** Acute Cystitis Symptom Score, cystitis, patient-reported outcome, questionnaire, women

## Abstract

The Acute Cystitis Symptom Score (ACSS) is a patient self-reporting questionnaire for the clinical diagnosis and patient-reported outcome (PRO) in women with acute uncomplicated cystitis (AC). The aim of the current study (part II) is the clinical validation of the Greek ACSS questionnaire. After linguistic validation according to internationally accepted guidelines and cognitive assessment (part I), the clinical validation was performed by using the Greek ACSS study version in 92 evaluable female participants including 53 patients with symptoms suspicious of AC and 39 controls. The clinical outcome using the ACSS questionnaire at different points in time after the start of treatment was demonstrated as well. The age (mean ± SD) of the 53 patients (44.7 ± 17.0 years) and 39 controls (49.3 ± 15.9 years) and their additional conditions at baseline visits, such as menstruation, premenstrual syndrome, pregnancy, menopause, diabetes mellitus, were comparable. There was, however, a significant difference (*p* < 0.001) between patients and controls at baseline visit regarding sum score of the ACSS domains, such as typical symptoms and quality of life. The clinical outcome of up to 7 days showed a fast reduction of the symptom scores and improvement of quality of life. The optimal thresholds for the patient-reported outcome of successful therapy could be established. The linguistically and clinically validated Greek ACSS questionnaire can now be used for clinical or epidemiological studies and also for patients’ self-diagnosis of AC and as a PRO measure tool.

## 1. Introduction

Acute uncomplicated cystitis (AC) is the most widespread inflammation of the bladder in women. It is caused predominantly by bacterial infections. The manifestation of AC as a response to certain medications, radiation therapy or potential irritants, such as feminine hygiene spray, spermicidal jellies or long-term use of a catheter is less frequent [1].

The majority of the patients with AC are immunocompetent women of childbearing age who have no comorbidities or urologic abnormalities. Menopausal women, however, are more prone to develop AC [2,3]. The onset of AC is painful and annoying, it has a characteristic symptomatology that includes urinary frequency, urgency and dysuria although unusual symptoms such as back pain and fatigue may also be present. Physical examination is usually normal, except for suprapubic tenderness in some cases.

A urinalysis is recommended to confirm the diagnosis. Urine cultures are recommended in women with suspected pyelonephritis, women with symptoms that do not resolve or that recur within two to four weeks after completion of the therapy, and in women who present with atypical symptoms [4].

Despite the frequency of AC, there are errors frequently committed and discrepancies unresolved. Even though there are general guidelines concerning diagnosis and classification, a wide variation of approaches exists in clinical practice [5,6,7]. Approximately 50% of patients with AC will recover without treatment; however, if left untreated, some patients with AC may progress to develop a recurrent infection, pyelonephritis and rarely renal failure [3]. In some patients, symptoms persist despite bacterial eradication, and it is not clear whether medications other than antibiotics are necessary to relieve patients from bothersome symptoms. While imaging procedures are not routinely required, it may be important to recognise potential complicating factors including anatomical and functional abnormalities of the urinary tract for certain patients [4,8].

For reasons stated above, AC must be carefully evaluated either individually or collectively. Well-drawn disease-specific questionnaires are easy-to-handle self-evaluation tools and may serve as appropriate smart instruments for gathering information from a large population of people within a short time. For this reason, they should be designed in such a way that data can be collected with the most optimal accuracy and the results are interpretable and generalisable. The Acute Cystitis Symptom Score (ACSS) was properly designed as an evaluation tool for AC. It is a simple and standardised self-reporting questionnaire assessing typical and differential symptoms, burden of symptoms, quality of life and evaluating possible changes in subjective well-being after treatment. It was originally developed in the Uzbek and Russian languages and has demonstrated high reliability and validity [9,10]. The present study was aimed to develop a Greek version of the ACSS questionnaire for clinical research, diagnosis of primary AC and monitoring the outcome and effectiveness of the treatment.

## 2. Results

A total of 92 evaluable participants were finally included: 53 with an acute symptomatic episode of cystitis (patients) and 39 visiting the physician for any other reason than UTI (controls). The clinical diagnosis of AC was confirmed by the treating physician according to national and international guidelines. The mean age (mean, SD) of the study group was 46.7 (49.3, 15.9) years. Three of the respondents were Albanian, one was Georgian and the remaining were Greek. All of the questions of the Greek version of the ACSS and the possible answers were easily understandable to all surveyed respondents (Figure 1). The demographics of the patients are shown in Table 1. None of the patients or controls had suffered from any other bladder disease or injury including surgery.

In all 53 patients diagnosed by the treating physician with cystitis, urine culture was performed before the start of therapy, while 31/53 (58.4%) had bacteriuria of ≥10^5^ CFU /mL, 15/53 (28.3%) had 10^4^ CFU/mL, 7/53 (13.2%) had <10^4^ CFU/mL. In 81.1% *Escherichia coli* was found at different levels of bacteriuria (Table 2).

Detailed results on the prevalence of complaints of varying severity (none, mild, moderate, severe) at Day 0 (baseline) are presented in Table 3 for items in the ACSS’s ‘typical’, ‘differential’ and ‘QoL’ domains.

The severity of the majority of symptoms and their negative impact on the quality of life were significantly higher in the group of patients than in controls. Similar results were found for the summary scores of all three domains and the cumulative summary scores for the ‘typical’ and ‘QoL’ domains, as well as for the total score of the ACSS (Table 4, Figure 2).

In patients, there was, however, no correlation between the amount of bacteriuria and summary scores of the different ACSS domains (Figure 3).

Along with the good values of internal consistency for different approaches, Cronbach’s alpha coefficient showed especially high values for the ‘typical’ and ‘QoL’ domains (Table 5).

The largest AUC in the ROC-curve analysis for individual ACSS items, corresponding to the most optimal diagnostic ability, was found for ‘urinary urgency,’ which was also confirmed by the diagnostic odds ratios (Figure 4 and Figure 5a). However, a higher balance for diagnostic sensitivity and specificity was found for the cumulative score of the ‘typical’ and ‘QoL’ domains, followed by the summary score of the ‘typical’ and ‘QoL’ domains which were higher than the AUC of the individual typical symptoms (Figure 4).

The frequency of the complaints about the vaginal discharge of different severity did not significantly differ between patients and controls (Table 3). We also did not find a statistically significant increase or decrease in diagnostic odds ratio depending on the presence or severity of the complaints about the vaginal discharge (Figure 5b).

Using the data of the 53 patients and 39 controls at visit 1 and using three different thresholds for the summary scores of the ‘typical’ ACSS symptoms, such as ≥5, ≥6 and ≥7, the highest diagnostic specificity was shown by thresholds ≥6 and ≥7, but the highest sensitivity by threshold ≥6. The overall diagnostic odds ratio, however, was best for threshold ≥6 (Table 6). Therefore, the cut-off by a summary score of ≥6 for the ‘typical’ domain should also be used for the Greek version as suggested earlier [9,10,11,12]. If the data of the 53 patients at visit 1 and the data of the 21 patients at the end of therapy visit (4–7 days after therapy) are used alternatively for the threshold of the summary score of ≥6 of the ‘typical’ ACSS symptoms, the sensitivity and the positive predictive value were the same, but the other parameters were slightly lower (Table 6).

All patients (*n* = 53) were treated and followed up to 7 days or shorter if they became asymptomatic earlier. Antimicrobial therapy was prescribed to 44 (83.0%) of them, whereas non-antimicrobial therapy was prescribed to 4 (7.5%), and combination therapy to 5 (9.4%) (Table 1). The summary scores of the ‘typical’ and ‘QoL’ domains were reduced during the follow-up visits as well as their cumulative score accordingly. Nevertheless, the reduction in scores was relatively faster for the ‘typical’ domain than for ‘QoL’ (Figure 6). This is similarly reflected by the reduction of the overall assessment by the ‘dynamics’ domain in patients at the three follow-up visits (Figure 7).

By comparing the thresholds, it can be seen that in 21 patients followed up 4–7 days after therapy at Day 4 according to threshold A (‘dynamics’) 16 patients showed ‘success’ and 5 patients ‘non-success’ (‘dynamics’ >1) (Table 7, Figure 8). In contrast, if the thresholds B, D and E,—considering symptoms only—are used, the success rates were higher and did not match with the overall assessment. If the two ACSS domains, ‘typical’ and ‘QoL’, were combined, the results were again similar to the results obtained by the domain ‘dynamics’ (Table 7, Figure 8). Although all 21 patients followed up to 7 days complained no more about ‘typical’ symptoms, the QoL assessment and overall assessment by the patients was delayed a few days as compared to assessment of symptom severity alone. Therefore, for patient-reported-outcome (PRO) assessment not only the severity of symptoms but also the QoL assessments should be considered.

## 3. Discussion

After translation and cognitive assessment [13], the Greek version of the ACSS was now also tested clinically in 53 female patients diagnosed with AC by the treating physician according to national and international guidelines and compared with the results of 39 controls without a suspected diagnosis of UTI. As expected, the two groups, patients and controls, differed statistically significantly regarding prevalence and severity of typical symptoms of AC. For the Greek patients a sum score of ≥6 of the typical symptoms showed also the best diagnostic odds ratio as with other languages in which the ACSS was tested clinically as well [14,15,16,17,18,19]. Therefore, the ACSS could now also be used in multilingual studies, in which the ACSS was validated linguistically and also tested clinically.

Since also non-antibiotic therapeutic modalities of AC have been discussed and tested recently [20,21,22,23], the primary aim of such studies should be more focused on the clinical outcome as reported by the patient. The routinely asked overall clinical assessment by the patient and the investigator is by definition subjective and thus imprecise. If patients are asked, however, the same questions regarding symptoms and their severity and the quality of life at different points in time before and after treatment, each patient will involuntarily develop an internal standard for himself. Thus, a graduation for the patient-reported outcome (PRO) can be tested and developed. Therefore, the clinical validation of such a questionnaire should also include follow-up investigations at different points in time to demonstrate its range and flexibility.

By using different thresholds for success and non-success it was interesting to demonstrate in this study that only questions about prevalence and severity of symptoms showed better results somewhat earlier as questions for overall outcome or questions for quality of life. Such differences may become of interest if different treatment modalities are compared in double-blind, prospectively randomised clinical trials. As has been shown earlier [18] there was no correlation between summary scores of any of the ACSS domains and the amount of bacteriuria when entering the study, which renders the insistence on certain significant bacteriuria as an inclusion criterion irrelevant. Interestingly, we again did not find evidence that the presence of complaints about vaginal discharge plays a significant role in the rejection of the diagnosis of AC, which may be another confirmation of our previous hypothesis that the presence of vaginal discharge does not reduce the likelihood of the presence of AC in women [24].

The shortcomings of the study are (1) performed in a single centre, and (2) in a limited number of patients (*n* = 53) and controls (*n* = 39). For all typical symptoms and QoL items, however, a significant difference could be found between patients and controls. Nevertheless, confirmation of the results in a larger multicentre study would be desirable.

## 4. Materials and Methods

After the linguistic validation (part I) as described earlier [13], we proceeded to the clinical validation of the Greek version of the ACSS. The local ethical committee of the Tzaneio Hospital provided the ethical approval for the whole study (Επιτροπή Hθικής και Δεοντολογίας, Τζανείο ΓΝΠ 6195/05-05-2020, 5 May 2020).

### 4.1. Study Design

This study was designed as a prospective, observational cohort study. The Greek version of the ACSS questionnaire (Figure 1) was administered prior, during and in several cases after treatment for as long as patients reported symptoms up to a maximum of seven days.

### 4.2. Study Subjects

Female subjects with an acute symptomatic episode of cystitis and those visiting the physician for any other reason except UTI, consecutively attending the Emergency Department of the Tzaneio General Hospital of Piraeus were recruited to the study. Women suffering from conditions that influence bacterial virulence or host response (e.g., immunodeficiency, abnormalities of the urogenital system) and patients who received antibiotics or immunosuppressive treatment within 4 weeks prior to the study were excluded from the analysis.

Respondents were categorised into patients and controls based on the treating physician’s diagnosis established following international and local guidelines. The members of the Scientific Committee responsible for the data processing and analysis did not have access to this information until the completion of the recruitment process.

### 4.3. Study Tool

The ACSS questionnaire consists of two parts (diagnostic part-A and follow-up part-B), each part containing 18 items divided into 4 domains. The first domain (‘typical’ domain) composed of 6 items examines typical acute cystitis symptoms, the second (‘differential’ domain) composed of 4 items examines symptoms suggesting infection of adjacent or relative organs (e.g., urethra, vagina, kidney). The third domain (‘QoL’ domain), composed of 3 items examines the impact of the symptoms on quality of life. There are 4 ranking choices for each of the above 13 items to accurately measure the severity of symptoms. The last domain (‘additional’ domain) is composed of 5 additional questions (requiring simple ‘Yes/No’) to gather information on the presence of known diabetes mellitus and gynaecological profile of the patient such as menstruation and pregnancy, additional conditions that may require the appropriate adjustment of the treatment modality. The follow-up part-B of the ACSS also contains the ‘dynamics’ domain to assess the overall clinical outcome reported by the patient [9,10,11].

### 4.4. Statistical Analysis

#### 4.4.1. Data Processing

Since in our previous studies we found that the ‘typical’ and ‘QoL’ domains have the highest AC-specific values, we particularly decided to evaluate and highlight their combined diagnostic significance in the current study. For this purpose, we summed up the total scores of both domains and subjected the cumulated value to statistical analysis along with the scores of other domains of the ACSS.

The efficacy of the therapy was classified as ‘success’ or ‘non-success’ according to the patients’ overall self-esteem using the ‘dynamics’ domain of the follow-up part B of the ACSS in which a score of not higher than 1 was rated as ‘success’, and the other predefined thresholds as suggested earlier [12,15]. Further data processing, in general, did not differ from that as described earlier [14,15,16,17,18].

#### 4.4.2. Statistical Tests

Normality of distributions as well as linearity and homoscedasticity of data were tested visually (using histograms, normal Q-Q Plots, etc.,) and mathematically using Shapiro–Wilk and Levene’s tests [25,26].

Reliability of the ACSS and its domains was assessed via internal consistency of the items and represented using Cronbach’s coefficient alpha and the coefficient of split-half reliability [27].

Discriminative ability was assessed by comparing the scores of the respective item scores and total domain scores between patients and controls at the diagnostic baseline visit. Responsiveness of the ACSS domains was measured by comparing patients’ total scores on different domains of the ACSS at baseline and follow-up visits and the changes in scores of the ‘dynamics’ domain at various follow-up visits.

Diagnostic abilities of the domains and items of the ACSS were assessed by the measurement of sensitivity, specificity, positive and negative predictive values, diagnostic odds ratio (DOR), Youden’s index and ROC-curve analyses.

Numerical values were presented by the values of measures of central tendency (e.g., mean, median), distribution and dispersion (e.g., standard deviation, 95% confidence intervals, interquartile range). A comparative analysis of the independent continuous variables was performed using a two-sided independent sample *t*-test and paired *t*-test with the Welch correction in cases of inequality of variances [28]. Categorical variables were presented in proportions and compared with McNemar’s test [29].

Comparisons of ordinal and interval variables, as well as values of matched groups (e.g., total scores of patients on ACSS domains at baseline and follow-up visits), were performed using the Wilcoxon signed-rank test [30]. The strength of associations was assessed using Pearson’s product-moment correlation coefficient for numerical and interval variables [31]. Statistical significance was set at 0.05.

R v.3.5.2 with in-built and additional packages was used for the analysis and graphical representation of the results [32,33,34,35,36].

#### 4.4.3. Sample Size Calculation

A sample size calculation for the general clinical validation of the ACSS with a predefined power of the test of 0.9, and a type I error rate (α) of 0.05 resulted in a necessary minimum number of participants of 58 (29 patients vs. 29 controls). According to Tsang et al. [37], the recommended ratio for the validation of a questionnaire is 5 respondents per item (5:1). Since the ACSS contains 18 items, a total number of 90 respondents (patients and controls) was considered as appropriate.

## 5. Conclusions

Although the study was conducted in a single centre, it demonstrated significant differences in all typical symptoms and QoL between patients and controls. The linguistically and clinically validated Greek ACSS questionnaire can now be used for clinical or epidemiological studies and also for patient’s self-diagnosis of AC and as a PRO measure tool. Further larger multicentre studies are welcome for the confirmation of the results.

The role of vaginal discharge in the probability of AC in women needs to be reassessed.

## 6. Patents

The ACSS is copyrighted by the Certificate of Deposit of Intellectual Property in Fundamental Library of Academy of Sciences of the Republic of Uzbekistan, Tashkent (Registration number 2463; 26 August 2015) and the Certificate of the International Online Copyright Office, European Depository, Berlin, Germany (Nr. EU-01-000764; 21 October 2015). The rightsholders are Jakhongir Fatikhovich Alidjanov (Uzbekistan), Ozoda Takhirovna Alidjanova (Uzbekistan), Adrian Martin Erich Pilatz (Germany), Kurt Guenther Naber (Germany) and Florian Martin Erich Wagenlehner (Germany). The e-USQOLAT is copyrighted by the Authorship Certificate of the International Online Copyright Office, European Depository, Berlin, Germany (Nr. EC-01-001179; 18 May 2017) 19. Translations of the ACSS in other languages are available on the website: http://www.acss.world/downloads.html (accessed on 10 October 2021)

## Figures and Tables

**Figure 1 antibiotics-10-01253-f001:**
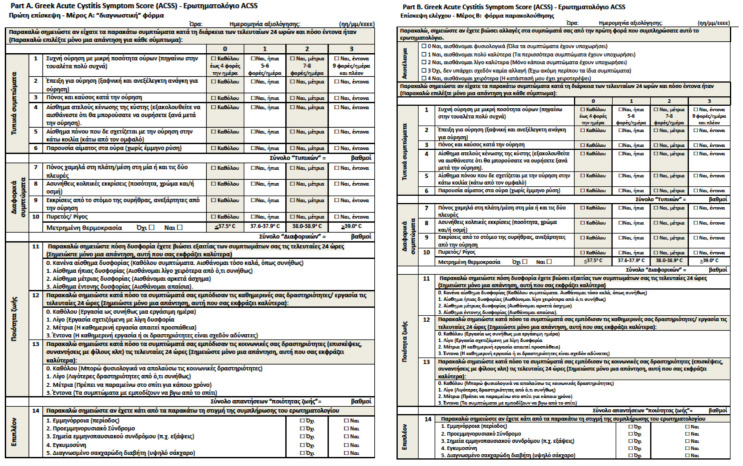
Greek Acute Cystitis Symptom Score (ACSS) questionnaire for visit 1 (Part A) and follow-up visits (Part B).

**Figure 2 antibiotics-10-01253-f002:**
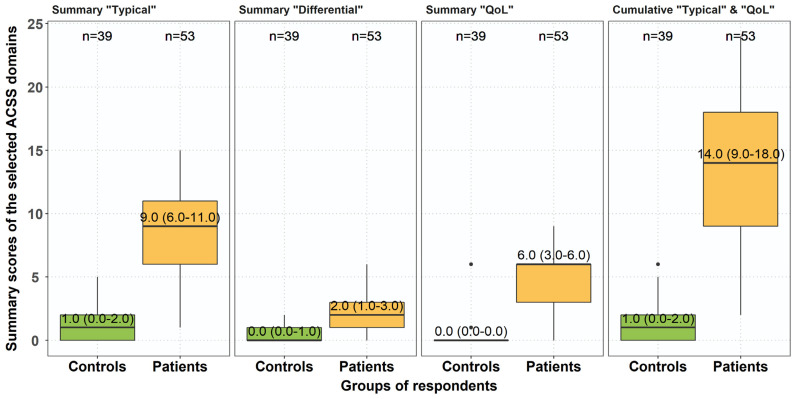
The difference in summary scores of the different ACSS domains between controls and patients. Boxplots (IQR, mean ± SD).

**Figure 3 antibiotics-10-01253-f003:**
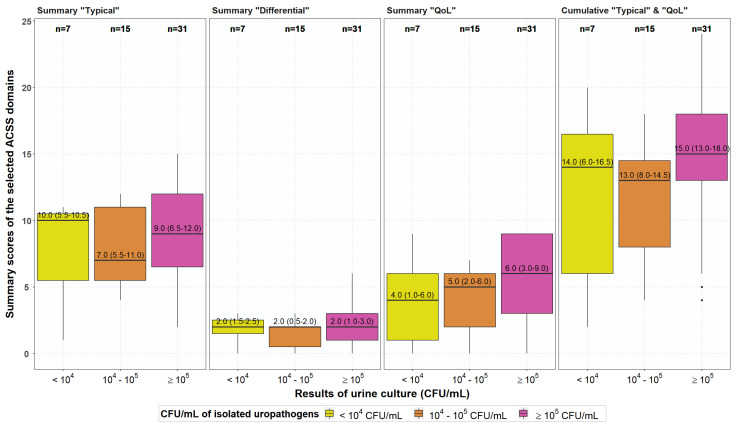
Correlation between summary scores of the different ACSS domains and cumulative scores in patients at baseline and different rates of bacteriuria.

**Figure 4 antibiotics-10-01253-f004:**
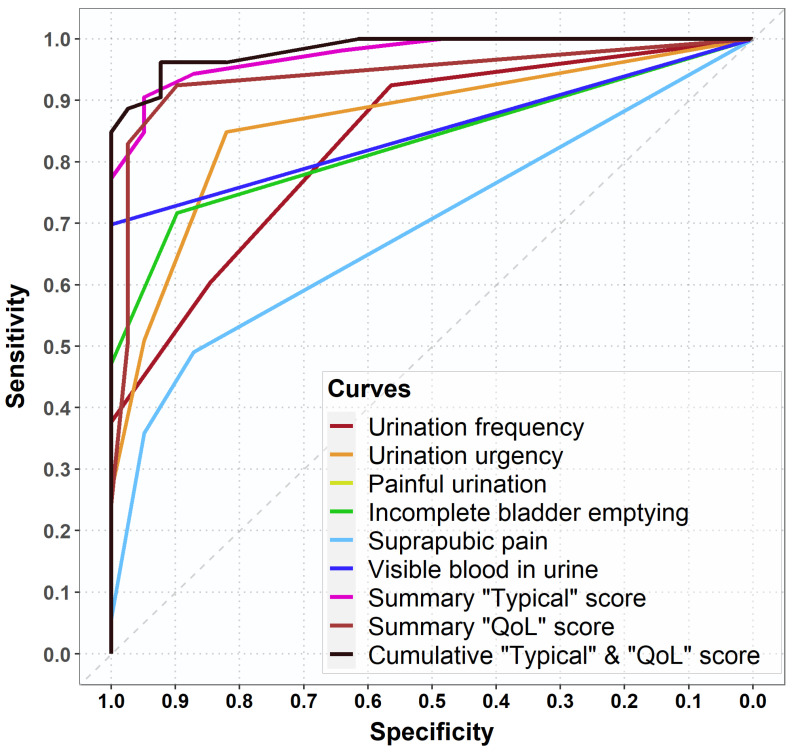
Receiver operating characteristic (ROC) curves for the ACSS typical symptoms, summary scores of the ‘typical’ and ‘QoL’ domains and the cumulative ‘typical’ and ‘Qol’ score at baseline (Day 0).

**Figure 5 antibiotics-10-01253-f005:**
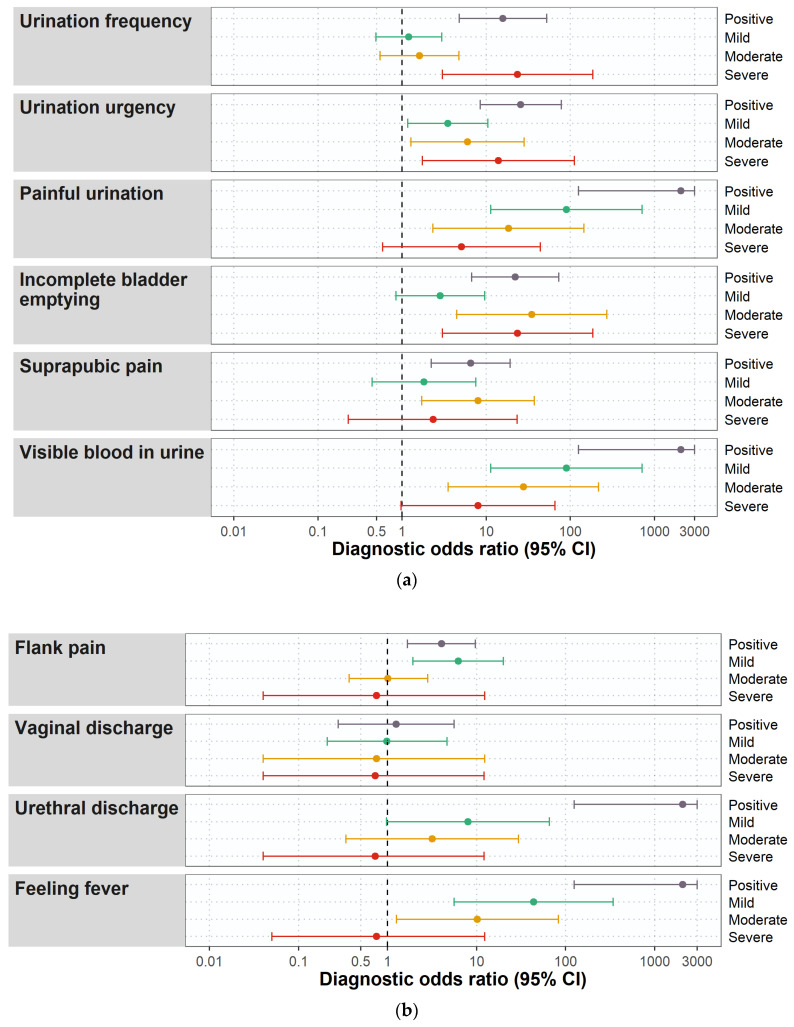
(**a**) Diagnostic odds ratios for the items of the ‘typical’ domain of the ACSS at baseline (Day 0). (**b**) Diagnostic odds ratios for the items of the ‘differential’ domain of the ACSS at baseline (Day 0).

**Figure 6 antibiotics-10-01253-f006:**
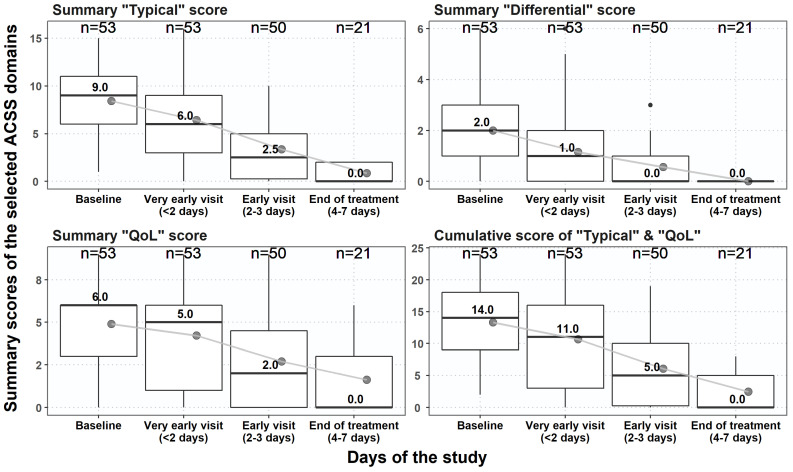
The course of changes in summary scores of the different ACSS domains and cumulative scores in patients at baseline and different days of follow-up visits. Boxplots (IQR, mean ± SD).

**Figure 7 antibiotics-10-01253-f007:**
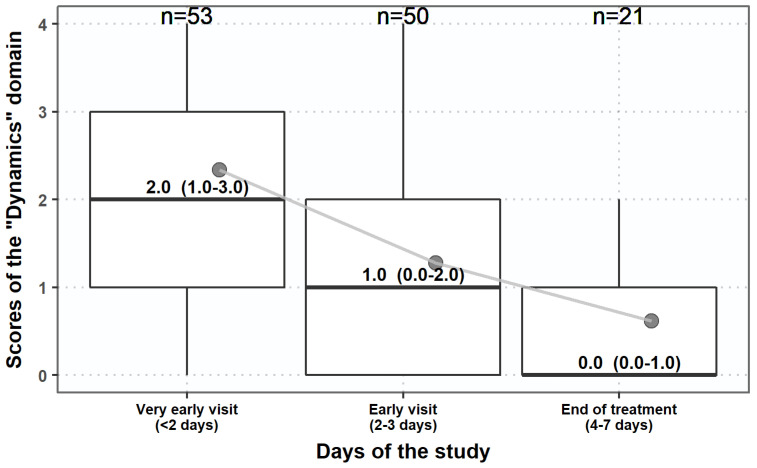
The course of changes in the ACSS ‘dynamics’ domain in patients at different days of follow-up visits. Boxplots (IQR, mean + SD).

**Figure 8 antibiotics-10-01253-f008:**
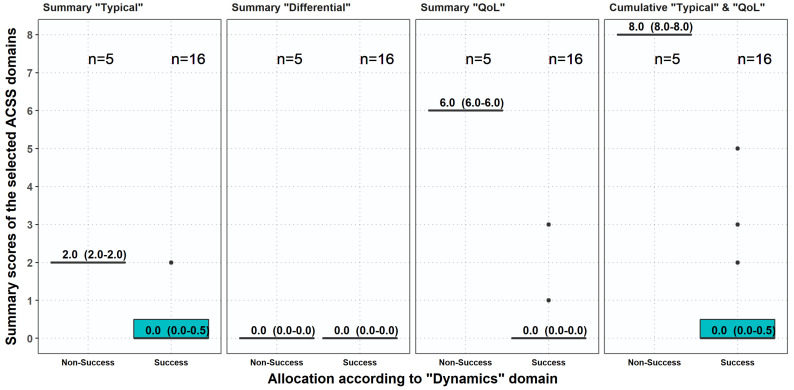
Comparison between ‘success’ and ‘non-success’ obtained in the ACSS domain ‘dynamics’ with the results obtained by summary scores in the ACSS domain ‘typical’, ‘QoL’ and by the cumulative summary of both domains. Boxplots (IQR, mean + SD).

**Table 1 antibiotics-10-01253-t001:** The demographic characteristics, additional conditions and the treatment modalities of the study population at baseline visit.

Parameter	Total Cohort	Controls	Patients	*p*-Value *
Number, *n* (%)	92 (100.0)	39 (100.0)	53 (100.0)	n.a.
Age, yr, mean (SD)	46.7 (16.6)	49.3 (15.9)	44.7 (17.0)	0.189
Employment				
Full-time, *n* (%)	16 (17.4)	n.a.	16 (30.2)	n.a.
Part-time, *n* (%)	17 (18.5)	n.a.	17 (32.1)	n.a.
Not working, *n* (%)	14 (15.2)	n.a.	14 (26.4)	n.a.
Retired, *n* (%)	6 (6.5)	n.a.	6 (11.3)	n.a.
Sexually active, *n* (%)	44 (47.8)	n.a.	44 (83.0)	n.a.
Symptomatic episodes of lower UTI within the last year				
No symptomatic episode, *n* (%)	47 (51.1)	n.a.	47 (88.7)	n.a.
One or two symptomatic episodes, *n* (%)	6 (6.5)	n.a.	6 (11.3)	n.a.
Received antimicrobial therapy within the last 3 months, *n* (%)	9 (9.8)	n.a.	9 (17.0)	n.a.
Urine culture performed, *n* (%)	53 (57.6)	0 (0.0)	53 (100.0)	n.a.
**Characteristics following ACSS’s ‘Additional’ domain**				
Cases with menstruation, *n* (%)	9 (9.8)	2 (5.1)	7 (13.2)	0.350
Cases with premenstrual symptoms, *n* (%)	9 (9.8)	5 (12.8)	4 (7.5)	0.627
Cases with symptoms of menopause, *n* (%)	10 (10.9)	3 (7.7)	7 (13.2)	0.616
Cases with pregnancy, *n* (%)	1 (1.0)	0 (0.0)	1 (1.9)	1.000
Cases with diabetes mellitus, *n* (%)	6 (6.5)	2 (5.1)	4 (7.5)	0.970
**Prescribed treatment modality**				
Antimicrobial therapy, *n* (%)	44 (47.8)	n.a.	44 (83.0)	n.a.
Non-antimicrobial therapy, *n* (%)	4 (4.3)	n.a.	4 (7.5)	n.a.
Combined therapy, *n* (%)	5 (5.4)	n.a.	5 (9.4)	n.a.

* Student *t*-test for numerical values, Wilcoxon–Mann Whitney test for ordinal/interval values and Pearson’s chi-square for categorical values. N.a.—not applicable.

**Table 2 antibiotics-10-01253-t002:** Microbiological results of urine cultures in 53 patients with AC.

Pathogens	Bacteriuria (Colony Forming Units (cfu) per mL)			
	≥10^5^/mL	10^4^/mL	10^3^/mL	10^2^/mL
*Escherichia coli*	28	13	1	1
*Proteus* sp.		2		
*Klebsiella* sp.	2			
*Enterococcus* sp.	1			
Staphylococcus CoN			4	1
Total *n* (%)	31 (58.4%)	15 (28.3%)	5 (9.4%)	2 (3.8%)

CoN—coagulase negative.

**Table 3 antibiotics-10-01253-t003:** Acute Cystitis Symptom Score (ACSS): Severity of symptoms in patients with AC and controls. Scores: none (0), mild (1), moderate (2), severe (3).

ACSS		Controls (*n* = 39)		Patients (*n* = 53)		*p*-Value
Domain	Item	Number	Percent	Number	Percent	
**Typical Symptoms**	1. Urinary frequency positive	17	43.6	49	92.4	<0.001
	None	22	56.4	4	7.6	<0.001
	Mild	11	28.2	17	32.1	0.694
	Moderate	6	15.4	12	22.6	0.391
	Severe	0	0.0	20	37.7	<0.001
	2. Urinary urgency positive	7	17.9	45	84.9	<0.001
	None	32	82.1	8	15.1	<0.001
	Mild	5	12.8	18	34.0	0.015
	Moderate	2	5.1	13	24.6	0.007
	Severe	0	0.0	14	26.4	<0.001
	3. Dysuria positive	0	0.0	37	69.8	<0.001
	None	39	100.0	16	30.2	<0.001
	Mild	0	0.0	20	37.7	<0.001
	Moderate	0	0.0	11	20.8	<0.001
	Severe	0	0.0	6	11.3	<0.001
	4. Incomplete bladder emptying positive	4	10.3	38	71.7	<0.001
	None	35	89.7	15	28.3	<0.001
	Mild	4	10.3	13	24.5	0.068
	Moderate	0	0.0	5	9.4	0.024
	Severe	0	0.0	20	37.7	<0.001
	5. Suprapubic pain positive	5	12.8	26	49.1	<0.001
	None	34	87.2	27	50.9	<0.001
	Mild	3	7.7	7	13.2	0.407
	Moderate	2	5.1	16	30.2	<0.001
	Severe	0	0.0	3	5.7	0.083
	6. Visible blood in urine positive	0	0.0	37	69.8	<0.001
	None	39	100.0	16	30.2	<0.001
	Mild	0	0.0	15	28.3	<0.001
	Moderate	0	0.0	13	24.5	<0.001
	Severe	0	0.0	9	17.0	<0.001
**Differential Symptoms**	7. Flank pain positive	12	30.8	34	64.2	0.001
	None	27	69.2	19	35.8	0.001
	Mild	4	10.3	22	41.5	<0.001
	Moderate	8	20.5	11	20.8	0.978
	Severe	0	0.0	1	1.9	0.322
	8. Vaginal discharge positive	3	7.7	5	9.4	0.773
	None	36	92.3	48	90.6	0.773
	Mild	3	7.7	4	7.6	0.980
	Moderate	0	0.0	1	1.9	0.322
	Severe	0	0.0	0	0.0	n.a.
	9. Urethral discharge positive	0	0.0	9	17.0	0.002
	None	39	100.0	44	83.0	0.002
	Mild	0	0.0	5	9.4	0.024
	Moderate	0	0.0	4	7.6	0.044
	Severe	0	0.0	0	0.0	n.a.
	10. Feeling fever positive	0	0.0	28	52.8	<0.001
	None	39	100.0	25	47.2	<0.001
	Mild	0	0.0	17	32.1	<0.001
	Moderate	0	0.0	10	18.9	0.001
	Severe	0	0.0	1	1.9	0.322
**Quality of Life**	11. General discomfort positive	2	5.1	49	92.5	<0.001
	None	37	94.8	4	7.6	<0.001
	Mild	1	2.6	9	17.0	0.015
	Moderate	1	2.6	17	32.1	<0.001
	Severe	0	0	23	43.1	<0.001
	12. Impact on everyday activity positive	1	2.6	31	58.5	<0.001
	None	38	97.4	22	41.5	<0.001
	Mild	1	2.6	5	9.4	0.024
	Moderate	0	0.0	12	22.6	0.002
	Severe	0	0.0	14	26.4	<0.001
	13. Impact on social life positive	3	7.7	38	71.7	<0.001
	None	36	92.3	15	28.3	<0.001
	Mild	2	5.1	12	22.6	0.012
	Moderate	3	2.6	14	26.4	<0.001
	Severe	0	0.0	12	22.6	<0.001

**Table 4 antibiotics-10-01253-t004:** Summary scores of the different domains of the ACSS at baseline (Day 0).

Summary Scores of the Different ACSS Domains	Total Cohort	Controls	Patients	*p*-Value *
Summary ‘Typical’ score, median (IQR)	5.0 (1.0–9.3)	1.0 (0.0–2.0)	9.0 (6.0–11.0)	<0.001
Summary ‘Differential’ score, median (IQR)	1.0 (0.0–2.0)	0.0 (0.0–1.0)	2.0 (1.0–3.0)	<0.001
Summary ‘QoL’ score, median (IQR)	1.0 (0.0–6.0)	0.0 (0.0–0.0)	6.0 (3.0–6.0)	<0.001
Cumulative ‘Typical’ and ‘QoL’ score, median (IQR)	6.0 (1.0–15.0)	1.0 (0.0–2.0)	14.0 (9.0–18.0)	<0.001
Cumulative score of the entire ACSS, median (IQR)	7.5 (2.0–17.0)	2.0 (0.0–3.0)	15.0 (12.0–20.0)	<0.001

* Student *t*-test.

**Table 5 antibiotics-10-01253-t005:** Cronbach’s alpha for internal consistency.

The ACSS Domain	Patients and Controls		Patients at Baseline and Follow-up Assessments		*p*-Value *
	Cronbach’s Alpha [95%CI]	Split-Half [95%CI]	Cronbach’s Alpha [95%CI]	Split-Half [95%CI]	
Typical	0.81 [0.75; 0.87]	0.83 [0.64; 0.90]	0.81 [0.77; 0.85]	0.83 [0.63; 0.92]	0.980
Differential	0.40 [0.22; 0.58]	0.33 [0.25; 0.62]	0.57 [0.48; 0.66]	0.51 [0.49; 0.69]	0.061
QoL	0.91 [0.88; 0.94]	0.81 [0.80; 0.81]	0.89 [0.86; 0.92]	0.77 [0.75; 0.78]	0.701
Cumulative ‘Typical’ and ‘QoL’	0.89 [0.86; 0.93]	0.89 [0.78; 0.94]	0.89 [0.86; 0.91]	0.89 [0.76; 0.95]	0.798
Entire ACSS	0.89 [0.86; 0.92]	0.89 [0.77; 0.94]	0.88 [0.86; 0.91]	0.89 [0.76; 0.94]	0.870

* Chi-square test of comparison between Cronbach’s alpha coefficients of different approaches.

**Table 6 antibiotics-10-01253-t006:** Comparison of three different cut-off points using summary scores of the ‘typical’ ACSS symptoms for clinical diagnosis of AC in patients at visit 1 and controls and * alternatively in patients at visit 1 and Day 4 to 7 (end of therapy visit).

Summary ‘Typical’ Score	5 and Higher	6 and Higher	7 and Higher	6 and Higher *
Sensitivity	0.74 (0.61; 0.84)	0.77 (0.64; 0.87)	0.70 (0.56; 0.82)	0.77 (0.64; 0.88)
Specificity	0.95 (0.83; 0.99	0.98 (0.87; 1.00)	0.98 (0.87; 1.00)	0.95 (0.77; 1.00)
Positive predictive value	0.96 (0.85; 0.99)	0.98 (0.87; 1.00)	0.97 (0.86; 1.00)	0.98 (0.87; 1.00)
Negative predictive value	0.70 (0.56; 0.82)	0.76 (0.63; 0.87)	0.71 (0.57; 0.82)	0.64 (0.45; 0.80)
Diagnostic odds ratio	52.0 (11.2; 241.0)	133.3 (16.5; 1073.6)	90.2 (11.4; 714.5)	71.8 (8.7; 589.9)
Youden’s index	0.69 (0.44; 0.84)	0.75 (0.51; 0.88)	0.67 (0.42; 0.82)	0.73 (0.41; 0.88)
Correlation with PO	0.67 (0.55; 0.77)	0.67 (0.54; 0.76)	0.61 (0.47; 0.72)	0.70 (0.56; 0.80)
Area under the ROC-curve	0.83 (0.76; 0.90)	0.83 (0.77; 0.89)	0.81 (0.75; 0.87)	0.82 (0.73; 0.90)

**Table 7 antibiotics-10-01253-t007:** Patient-reported outcome (PRO) in 21 patients at the ‘end of therapy visit’ (days 4 to 7 after the start of the appropriate therapy). ‘Success’ in the ACSS domain ‘dynamics’ corresponds to a score with no item >1, which is compared with other measures of ‘success’ according to different thresholds.

Type	Thresholds for Clinical Success	Success	Non-Success
A	Dynamics, no item >1	16	5
B	Sum score of typical domain ≤ 5 scores, no item > 1	21	0
C	Sum score of typical domain < 5 scores, no item > 1 and no item of QoL > 1	16	5
D	Sum score of the 4 FDA symptoms ≤ 4, no item > 1	21	0
E	Sum score of the 3 EMA symptoms ≤ 3, no item> 1	21	0

4 FDA symptoms: urinary frequency, urinary urgency, dysuria, suprapubic pain; 3 EMA symptoms: urinary frequency, urinary urgency, dysuria. Type B-E: and no visible blood in urine.

## Data Availability

Not Applied.

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
