# Peer review of "Clinical Validation of the Greek Version of the Acute Cystitis Symptom Score (ACSS)—Part II"

_antibiotics, 2021, doi:10.3390/antibiotics10101253_

Round 1

Reviewer 1 Report

This work reports the clinical validation of the Greek Acute Cystitis Symptom Score (ACSS)  questionnaire. Generally, the results could be valuable for physicians and other researchers in this field. Some comments were suggested as follows.
1. Some critical references are missing in various statements:
“Physical examination is usually normal, except for suprapubic tenderness in some cases. In rare cases, palpatory and percussive tenderness in costovertebral angle may be present as well.”
“some patients with AC may progress to develop a recurrent infection, pyelonephritis, and rarely renal failure”
2. Table 1: I recommend the information of bladder injury and surgery history should be added in this table for Patients and Controls.
3. One limitation of this work is it is a single-center design. It is not sure whether the study is representative of the Greek women population due to the high variability in subjects born or permanently resident in other areas of the country.

Author Response

Dear Reviewers,

We would like to thank the reviewers for their reviews which we have followed as much as possible.

Reviewer 1.

  1. English language and style are fine/minor spell check required

Response: The manuscript was now checked by a professional English translator

  1. Introduction can be improved.

Response: We took out one sentence because we could not find a good reference and we added one reference for another sentence as requested (see below)

  1. Conclusions can be improved.

Response: We added in the Conclusions: “Although the study was performed in a single centre with limited numbers of patients and controls, significant differences were shown for all typical symptoms and QoL issues between patients and controls. Despite confirmation of the results in a larger multicentre study would be desirable,” the linguistically and clinically validated Greek ACSS questionnaire can now be used……

  1. Some critical references are missing in various statements:

4a. “Physical examination is usually normal, except for suprapubic tenderness in some cases. In rare cases, palpatory and percussive tenderness in costovertebral angle may be present as well.”

Response: Thank you for your comment. There is no good reference for the finding that “In rare cases, palpatory and percussive tenderness in costovertebral angle may be present as well.” Therefore we took out this sentence.

4b. “Approximately 50% of patients with AC will recover without treatment; however, if left untreated, some patients with AC may progress to develop a recurrent infection, pyelonephritis, and rarely renal failure.” Reference is missing.

Response: We included the reference, which is reference [3].

  1. Table 1: I recommend the information of bladder injury and surgery history should be added in this table for Patients and Controls.

Response: After Table 1 is mentioned in the text of “Results” the following sentence was included: “None of the patients or controls had suffered from any other bladder disease or injury including surgery.”

  1. One limitation of this work is it is a single-center design. It is not sure whether the study is representative of the Greek women population due to the high variability in subjects born or permanently resident in other areas of the country.

Response: We added at the end of Discussion: “The shortcomings of the study are 1) performed in a single centre, and 2) in a limited number of patients (n=53) and controls (n=39). For all typical symptoms and QoL items, however, a significant difference could be found between patients and controls. Nevertheless, confirmation of the results in a larger multicenter study would be desirable.” And we added in Material and Methods a short paragraph “4.4.3. Sample size calculation” (see below) to justify the numbers included.

Reviewer 2

  1. English language and style are fine/minor spell check required.

Response: The manuscript was now checked by a professional English translator

  1. Describing of methods could be improved.

Response: see below: Sample size calculation and shortcomings.

  1. In methodology section: The number of patients is 53 while the controls are 39. Is the presence of 53 patients good enough to validate the questionnaire? And why 39 controls?

Response: Why these number? At the end of “Materials” we added a paragraph:

4.4.3. Sample size calculation

For sample size calculation with a predefined power of the test of 0.9, and a Type I error rate of 0.05 a minimum number of participants of 29 per arm (29 Patients vs. 29 Controls) was necessary. According to Tsang et al [37], the recommended ratio for the validation of a questionnaire is 5 respondents per item (5:1). Since the ACSS contains 18 items, a total number of 90 respondents (patients plus controls) were needed. Both calculations were considered.

We also added at the end of Discussion: “The shortcomings of the study are 1) performed in a single centre, and 2) in a limited number of patients (n=53) and controls (n=39). For all typical symptoms and QoL items, however, a significant difference could be found between patients and controls. Nevertheless, confirmation of the results in a larger multicenter study would be desirable.”

All changes incl. linguistic changes only were marked yellow.

We hope, that the manuscript will now be ready for publication.

Reviewer 2 Report

Dear Prof. NABER

First of all it is a great honor and pleasure for me to review your extraordinary study, as always. Your excellent papers help us to solve the unsolved problems in the fields of diagnostics and treatment of UTIs.

I read your invaluable manuscript. There is just a question to explain for the readers.

In methodology section: The number of patients is 53 while the controls are 39. Is the presence of 53 patients good enough to validate the questionnaire? And why 39 controls?

Thank you very much
Best Regards

Author Response

(The authors gave the same response as above.)
